# YAP inhibits HCMV replication by impairing STING-mediated nuclear transport of the viral genome

Ju Hyun Lee[1], Mookwang Kwon[2], Woo Young Lim[1], Chae Rin Yoo[1], Youngik Yoon[1], Dasol Han[2], Jin-Hyun Ahn[3], Keejung Yoon [1,2]*

1 Department of Biopharmaceutical Convergence, Sungkyunkwan University, Suwon, South Korea,
2 College of Biotechnology and Bioengineering, Sungkyunkwan University, Suwon, South Korea,
3 Department of Molecular Cell Biology, Sungkyunkwan University School of Medicine, Sungkyunkwan University, Suwon, South Korea

* keejung@skku.edu

**Data Availability Statement:** All relevant data are within the manuscript and its Supporting information files.

## Abstract

YES-associated protein (YAP), a critical actor of the mammalian Hippo signaling pathway involved in diverse biological events, has gained increased recognition as a cellular factor regulated by viral infections, but very few studies have investigated their relationship vice versa. In this study, we show that YAP impairs HCMV replication as assessed by viral gene expression analysis and progeny assays, and that this inhibition occurs at the immediate-early stages of the viral life cycle, at the latest. Using YAP mutants lacking key functional domains and shRNA against TEAD, we show that the inhibitory effects of YAP on HCMV replication are nuclear localization- and TEAD cofactor-dependent. Quantitative real-time PCR (qPCR) and subcellular fractionation analyses reveal that YAP does not interfere with the viral entry process but inhibits transport of the HCMV genome into the nucleus. Most importantly, we show that the expression of stimulator of interferon genes (STING), recently identified as an important component for nuclear delivery of the herpesvirus genome, is severely downregulated by YAP at the level of gene transcription. The functional importance of STING is further confirmed by the observation that STING expression restores YAP-attenuated nuclear transport of the HCMV genome, viral gene expression, and progeny virus production. We also show that HCMV-upregulated YAP reduces expression of STING. Taken together, these findings indicate that YAP possesses both direct and indirect regulatory roles in HCMV replication at different infection stages.

## Author summary

Human cytomegalovirus (HCMV) infections in immunocompromised patients cause morbidity and mortality, and congenital infections can lead to severe neurodevelopmental deficits such as microcephaly. This study shows that YAP, a transcription factor renowned for its roles in diverse cellular events, potently inhibits HCMV viral gene expression and subsequent infectious progeny production. Furthermore, we find that these inhibitory

**Funding:** This research was supported by the National Research Foundation of Korea (NRF) funded by the Ministry of Science and ICT, the Republic of Korea (2021R1A2C1009018) (KY). The funders had no role in study design, data collection and analysis, decision to publish, or preparation of the manuscript.

effects of YAP on HCMV replication are due to inefficient delivery of the HCMV genome into the nuclei of the host cells, thus hindering a key early step of the viral life cycle. Importantly, we pinpoint that YAP reduces STING protein levels, contributing to the impaired nuclear transport of the HCMV genome. Taken together, our findings provide insight into the molecular features of YAP regulation of viral replication, and novel antiviral approaches against HCMV and other herpesvirus infections.

## Introduction

Human cytomegalovirus (HCMV) is a double-stranded DNA virus and a member of the Beta-herpesvirinae subfamily. HCMV is a ubiquitous virus with a seroprevalence rate of up to 90% in the adult population of the United States [1]. HCMV infection in healthy individuals is mostly asymptomatic, but it has been linked to a wide spectrum of illnesses in newborns and immunocompromised individuals. For example, fetal HCMV infection can cause serious neurodevelopmental disorders, including hearing loss, cerebral palsy, mental retardation, and microcephaly [2–4]. However, despite its clinical significance, the exact mechanism of HCMV replication is still not fully understood and strategies for the treatment of HCMV infection are in need of improvement.

The Hippo signaling pathway is an evolutionarily conserved signal transduction system well-known for its originally identified ability to control cell growth and organ size [5]. Although discovered relatively recently, the Hippo pathway has received great attention from many researchers due to its involvement in diverse biological processes. The mammalian Hippo pathway comprises mammalian sterile 20-like kinase 1 (MST1), large tumor suppressor 1/2 (LATS1/2) and Yes-associated protein (YAP), a transcription factor. YAP is the ultimate executor of the Hippo pathway, and its cytoplasm-nuclear shuttling and proteasomal degradation are regulated by the kinase activity of LATS1/2 [6]. Because YAP lacks DNA-binding activity, it must interact with other DNA-binding transcription cofactors to regulate target gene expression [7]. Among the cofactors, transcriptional enhanced associate domain (TEAD) family transcription factors are the most studied partners because they are required for YAP-regulated cell growth, which was the first identified function of YAP [8]. YAP also contains WW domains capable of association with proline-rich motif (represented by the sequence PPXY)-containing transcriptional coactivators such as Runx, AP2, C/EBP and c-Jun [9–12].

Over the past decade, numerous studies have linked viral pathogenesis to YAP activity. For example, infection of Epstein–Barr virus (EBV), hepatitis B virus (HBV), hepatitis C virus (HCV), human papillomavirus (HPV), and Kaposi sarcoma-associated herpesvirus (KSHV) were reported to result in changes in YAP protein levels through various mechanisms such as activation of the YAP promoter or inhibition of YAP protein degradation [13–17]. However, relatively fewer studies have investigated the vice versa relationship; the effects of YAP on viral infection.

In this study, we show that YAP expression impairs HCMV gene expression and that this inhibition is due to reduced nuclear transport of the HCMV genome at the early steps of the viral life cycle. Furthermore, we show that YAP-reduced expression of stimulator of interferon genes (STING) is responsible for the impeded delivery of the HCMV viral genome, providing a detailed molecular mechanism by which YAP regulates HCMV replication.

## Results

### YAP expression impairs HCMV progeny production

To test the effects of YAP on HCMV replication, we first performed a progeny virus assay titrating infectious virion particles produced after one round of viral replication. Briefly, prior to HCMV infection, human foreskin fibroblast (HFF) cells were transduced with a retroviral vector expressing YAP and GFP bicistronically (Fig 1A). The cells were then infected with HCMV at a multiplicity of infection (MOI) of 0.1 or 0.5. Progeny viruses were harvested from 72 to 120 h post-infection (hpi) and titered by a second round of infection of untransduced HFF cells followed by immunostaining using an anti-IE1 antibody (Fig 1B). As shown in Fig 1C–1F, infectious HCMV virions were detected from 72 hpi control samples at both MOI of 0.1 and 0.5, and their numbers increased with time. However, YAP expression in host cells was found to cause a great reduction in infectious virion production regardless of MOI or harvest time. Retroviral vector-mediated overexpression systems may produce transgene expression at a level higher than physiological levels. Based on the fact that YAP activation is achieved by reduced activity of LATS1/2, a direct YAP upstream kinase that induces YAP degradation during normal cellular processes [6], we mimicked these conditions by introducing a dominant negative form of the LATS1/2 gene (dnLATS1/2) into host cells and observed that dnLATS1/2 upregulated YAP protein levels, and shRNA vectors against YAP (shYAP) effectively abolished dnLATS1/2-induced YAP expression (Fig 1G and 1H). As shown in Fig 1I and 1J, expression of dnLATS1/2 potently reduced HCMV progeny production and additional expression of shYAP reversed the effects of dnLATS1/2, indicating that endogenously increased YAP is also sufficient to inhibit HCMV replication. Taken together, these results suggest that YAP has a strong negative impact on HCMV progeny virus production in host cells.

### HCMV viral gene expression is attenuated by YAP from the immediate-early stages of infection

Defects at any stages of the viral life cycle can ultimately lead to reduced progeny virus yield. To determine which step(s) of HCMV replication was affected by YAP, we performed quantitative real-time PCR (qPCR) analysis to measure the mRNA levels of HCMV viral genes. We found that mRNAs of all representative immediate-early (IE1), early (UL44 and UL83) and late (UL99 and UL108) phase-specific HCMV genes were significantly reduced in YAP-expressing host cells (Fig 2A). Western blot analysis confirmed the impaired HCMV gene expression at the protein levels (Fig 2B and 2C). Since HCMV immediate-early genes are required to ensure the transcription of later stage genes and thus progression into the early and late phases of the infectious cycle [18], these data suggest that YAP interference of HCMV replication occurs at the immediate-early stages of the viral life cycle, at the latest.

### TEAD-dependent transcriptional activity of YAP is important for its inhibitory effects on HCMV replication

YAP is well-known as a transcription factor acting in the nucleus, however, non-transactivational and non-nuclear roles of YAP have also recently been revealed in many cellular events. For example, cytoplasmic YAP can sequester β-catenin into the cytoplasm, resulting in the downregulation of Wnt/β-catenin signaling [19]. Therefore, we first tested if nuclear localization is required for the suppressive role of YAP in HCMV replication by using a YAP mutant lacking the PDZ-binding motif (YAP ΔPDZ) (Fig 3A), which is known to be an important domain for nuclear translocation of YAP proteins [20]. One of the most well-known functions

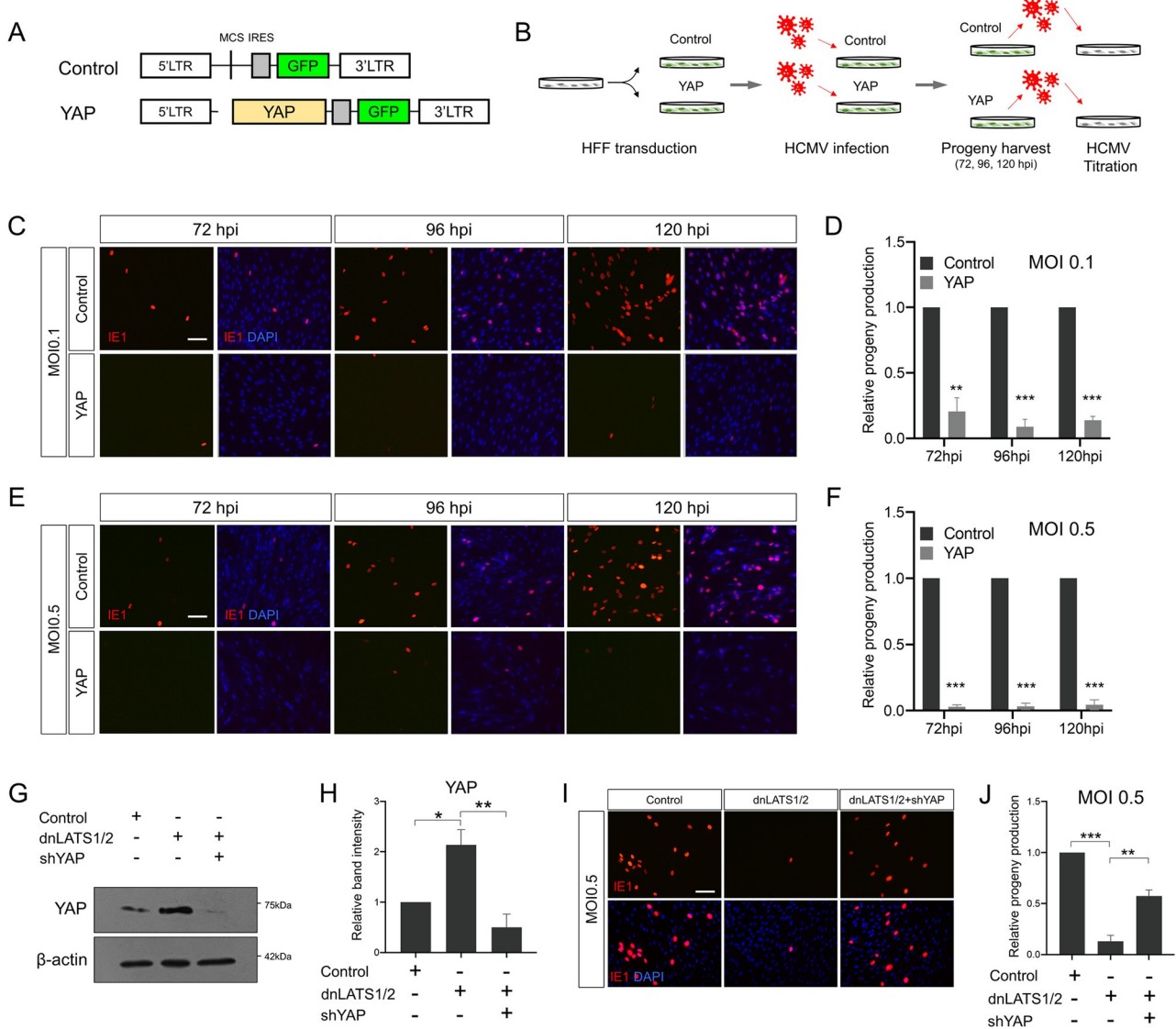

**Fig 1. HCMV replication is inhibited by YAP expression.** (A) Schematic representation of the retroviral vector used in this study. A retroviral vector which bicistronically expresses YAP and GFP using an internal ribosome entry site (IRES) was used to transduce HFF cells. As a negative control, retroviral vectors containing only the GFP gene were used. Therefore, GFP was used as a marker for transgene expression throughout the study. LTR, long terminal repeat; MCS, multicloning site. (B) HCMV progeny virus titration assay. 24 h prior to HCMV infection, HFF cells were transduced with a retroviral vector bicistronically expressing YAP and GFP. The cells were then infected with HCMV at an MOI of 0.1 or 0.5, and progeny viruses were harvested at 72, 96, and 120 hpi. Before infection for viral titration, all the MOI 0.5 samples were diluted to 1:10. Titration of progeny viruses collected from control- or YAP-transduced HFF cells infected with HCMV at an MOI of (C) 0.1 and (E) 0.5 at the indicated time points by anti-IE1 immunostaining (red). Cells were counterstained with DAPI to visualize nuclei (blue). (G) Western blot analysis for YAP proteins in HFF cells transduced with a dominant negative form of LATS1/2 (dnLATS1/2) together with or without shRNA specific to YAP (shYAP). (I) HFF cells were transduced with dnLATS1/2 and/or shYAP and then infected with HCMV at an MOI of 0.5. Progeny viruses were titered by IE1 immunostaining after HFF cell infection (red). (D, F, H, J) Quantification of (C, E, G, I), respectively. For (D, F, J), numbers of IE1⁺ cells in each control were set to 1. Scale bars, 100 μm. *n* = 3 biological replicates for each experiment. Error bars represent SEM. Student's *t*-test (for D, F), one-way ANOVA with Turkey's multiple comparison test (for H, J) was used to determine statistical significance. $^*P < 0.05$, $^{**}P < 0.01$ and $^{***}P < 0.001$.

of YAP is to increase cell proliferation. Thus, to rule out the possibility that cell proliferative activity affected HCMV replication, we performed an XTT assay under experimental HCMV treatment conditions which wait until host cell confluence reaches 100% before infection, and found no differences in cell proliferation between wild-type and mutant YAP-transduced

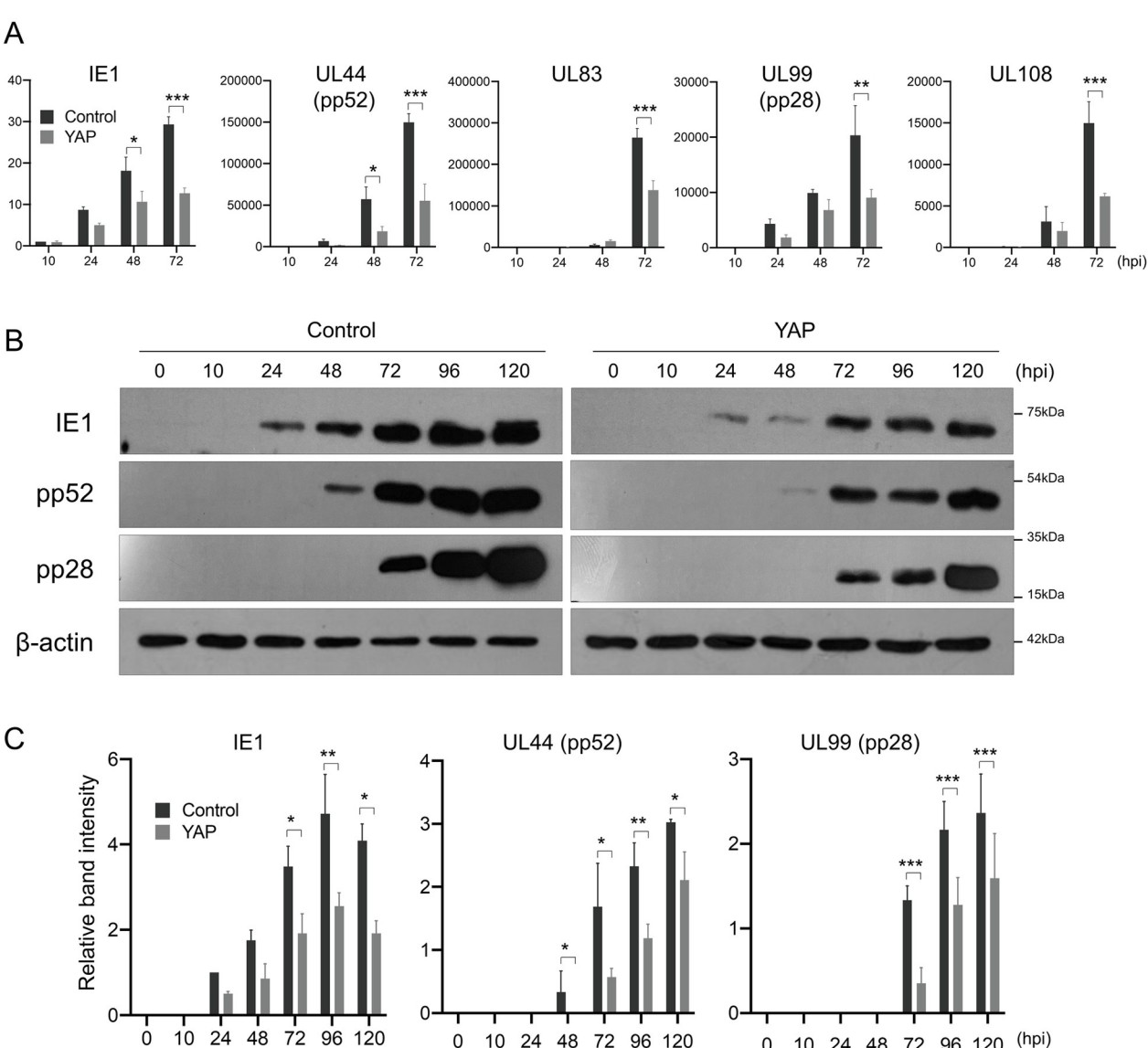

**Fig 2. HCMV immediate-early gene expression was reduced by YAP expression.** (A) qPCR analysis for representative immediate-early (IE1), early (UL44 and UL83) and late (UL99 and UL108) viral gene expression using YAP-transduced HFF cells infected with HCMV at an MOI of 0.5 and harvested at the indicated hpi. Each mRNA level was normalized to β-actin mRNA. (B) Western blot for HCMV IE1, pp52 and pp28 proteins using YAP-expressing HFF cells infected with HCMV at MOI of 0.5 and harvested at the indicated hpi. (C) Quantification of (B). Error bars represent SEM. $n$ = 3 biological replicates for each experiment. Two-way ANOVA with Sidak's multiple comparisons test was used to determine statistical significance. $^{*}P < 0.05$, $^{**}P < 0.01$ and $^{***}P < 0.001$.

samples (Fig 3B). As shown in Fig 3C and 3D, expression of YAP ΔPDZ did not affect HCMV progeny production. Furthermore, the YAP S94A mutant, which lacks binding affinity to the TEAD family transcription factors [8], also failed to reduce infectious viral particle production. In addition, we tested three YAP WW domain mutants in which the amino acid residues W199QDP202 and W258LDP261 were individually or simultaneously mutated to A199QDA202 and A258LDA261 to compromise the WW1 and WW2 domain functions, respectively (Fig 3A) [21]. Unlike the YAP ΔPDZ and YAP S94A mutants, a progeny virus assay showed that all WW1/2 mutants behaved like wild-type YAP (Fig 3E), suggesting that

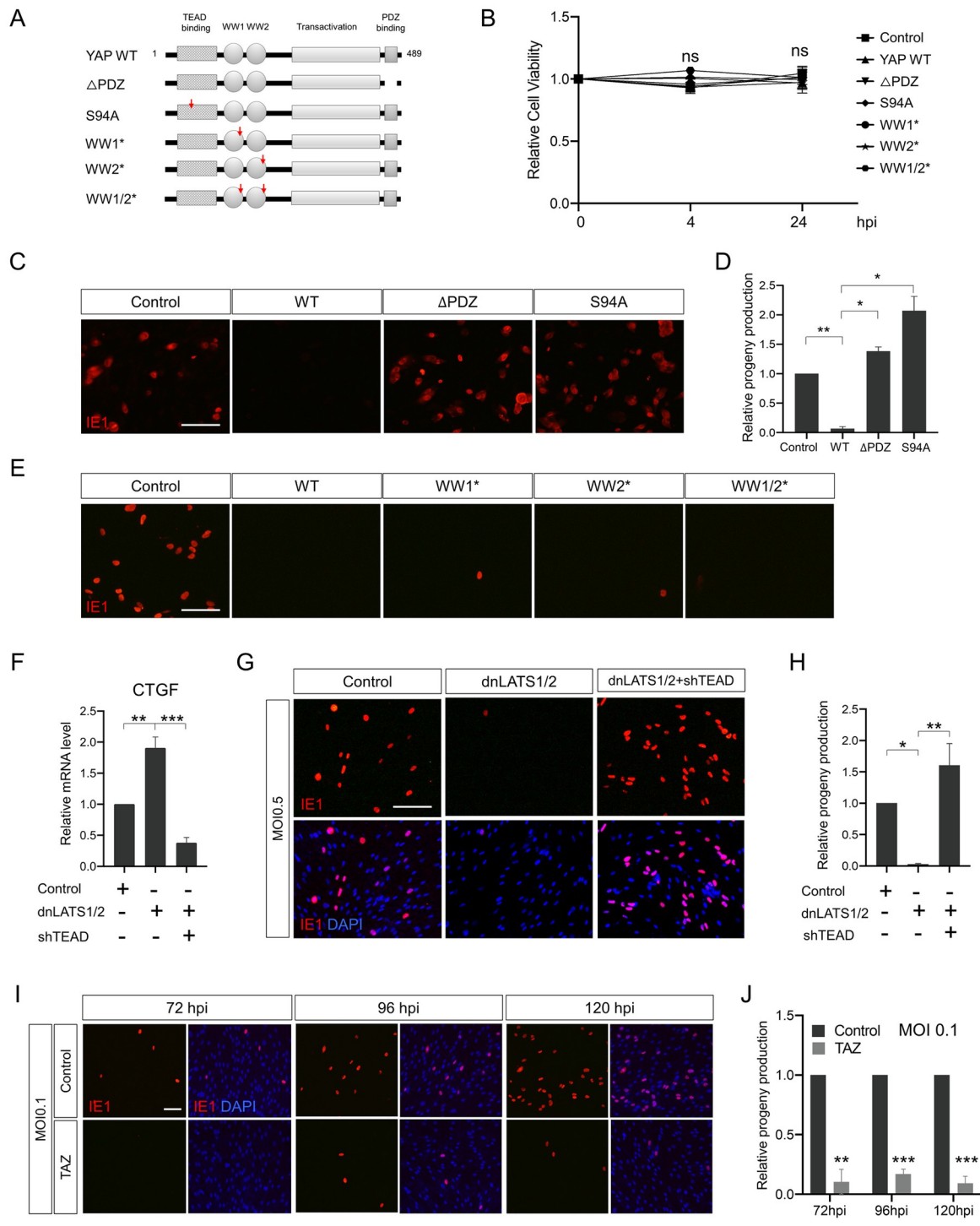

**Fig 3. YAP inhibition of HCMV progeny virus production is transcriptional activity-dependent.** (A) Schematic representation of wild-type and mutant YAP genes used in this study. (B) HFF cells were transduced with a retroviral vector expressing YAP genes and harvested for an XTT assay 4 and 24 h after the cell confluence reached 100%. Titration of progeny viruses harvested at 120 hpi from (C) YAP ΔPDZ or YAP S94A-transduced cells, and (E) YAP WW domain mutant-transduced cells infected with HCMV at an MOI of 0.5. (F) qPCR analysis of CTGF gene expression using dnLATS1/2 and TEAD shRNA (shTEAD)-transduced HFF cells at 2 days post-transduction. (G) HFF cells were transduced with dnLATS1/2 together with or without shTEAD-expressing retroviral vectors, and then infected with HCMV at an MOI of 0.5. Progeny viruses were titered by IE1 immunostaining after HFF cell infection (red). (I) Effects of TAZ expression on HCMV progeny virus production. TAZ-transduced HFF cells were infected with HCMV, and progeny viruses were harvested at 72, 96, and 120 hpi. Titration of progeny viruses was performed by anti-IE1 immunostaining (red). Cells were

counterstained with DAPI to visualize nuclei (blue). (D, H, J) Quantification of (C, G, I), respectively. Scale bars, 100 μm. *n* = 3 biological replicates for each experiment except for (B, *n* = 5 biological replicates). Error bars represent SEM. One-way ANOVA with Turkey's multiple comparison (for D, F, H), and Student's *t*-test (for J) were used to determine statistical significance. $^*P < 0.05$, $^{**}P < 0.01$ and $^{***}P < 0.001$.

WW domain-interacting transcriptional co-activators such as AP2, p73 and RUNX2 are not involved in YAP-regulated HCMV replication. Furthermore, to test the association of TEAD in YAP functions more directly, we employed TEAD shRNA (shTEAD) which completely diminished dnLATS1/2-induced expression of CTGF, a well-known YAP target gene [8], to levels lower than the control (Fig 3F). As shown in Fig 3G and 3H, dnLATS1/2-suppressed HCMV progeny production was dramatically rescued by shTEAD expression.

Since the property of YAP as a nuclear transcriptional factor was revealed to be important for the suppression of HCMV replication, we questioned whether the YAP paralog, transcriptional co-activator with PDZ-binding motif (TAZ), has a similar function. As shown in Fig 3I and 3J, TAZ also greatly reduced infectious HCMV virion production. These data indicate that YAP inhibition of HCMV replication is nuclear localization- and TEAD cofactor-dependent, and that TAZ, the other mammalian Yorkie homolog, also can impede HCMV replication.

## YAP inhibits nuclear transport of the HCMV genome by repressing STING gene expression

Reduced IE1 mRNA levels suggested that YAP affects very early stages of the HCMV replication cycle and that YAP may exert its function before viral gene expression begins. Therefore, we initially tested the first step of viral infection; the viral entry process. HCMV-added HFF cells were kept at 4°C for 1 h for attachment of virion particles to the host cell surface to synchronize infection and incubated at 37°C for an additional 1 h to allow viral particles to enter the cells. The viral genome that had just entered the cell was then analyzed by qPCR to detect the genomic IE1 sequence. As shown in Fig 4A, the relative amounts of viral DNA prepared within 1 h post-entry were not changed by YAP expression, indicating that YAP has no effect on HCMV entry into host cells. We then moved to the next stage of viral infection; nuclear delivery of the HCMV genome. We infected HFF cells with the same viral load of HCMV and harvested them at 4 hpi for subcellular fractionation. qPCR analysis showed that viral genome levels assessed by genomic IE1 amplification were significantly lower in the nuclear fraction of YAP-expressing cells than in that of control cells (Fig 4B and 4C). These data suggest that YAP-induced impairment of HCMV replication is due to inefficient import of the viral genome into the nucleus.

Nuclear delivery of the HCMV genome can be regulated in diverse ways. However, since the transcriptional activity of YAP was revealed to be critical for its inhibitory effect on HCMV replication, we hypothesized that YAP-blocked nuclear delivery of the HCMV genome is due to the abnormal expression of a gene responsible for viral nuclear import. We performed qPCR analysis and searched for a HCMV nuclear transport-related gene whose expression levels were altered by YAP expression. Stimulator of interferon genes (STING) was originally identified as a downstream factor of cyclic GMP–AMP (cGAMP) synthase (cGAS) that induces the innate immune response against pathogen infection by triggering the TBK1/IRF3 signaling cascade [22], and was recently discovered to facilitate nuclear import of the HCMV genome [23]. We found that YAP expression significantly reduced the amounts of STING protein (Fig 4D and 4E) and that this reduction occurred at the mRNA level (Fig 4F). Interestingly, STING mRNA was increased upon HCMV infection, and this increase was also effectively reduced by YAP expression (Fig 4G). Typically, STING proteins form foci in the

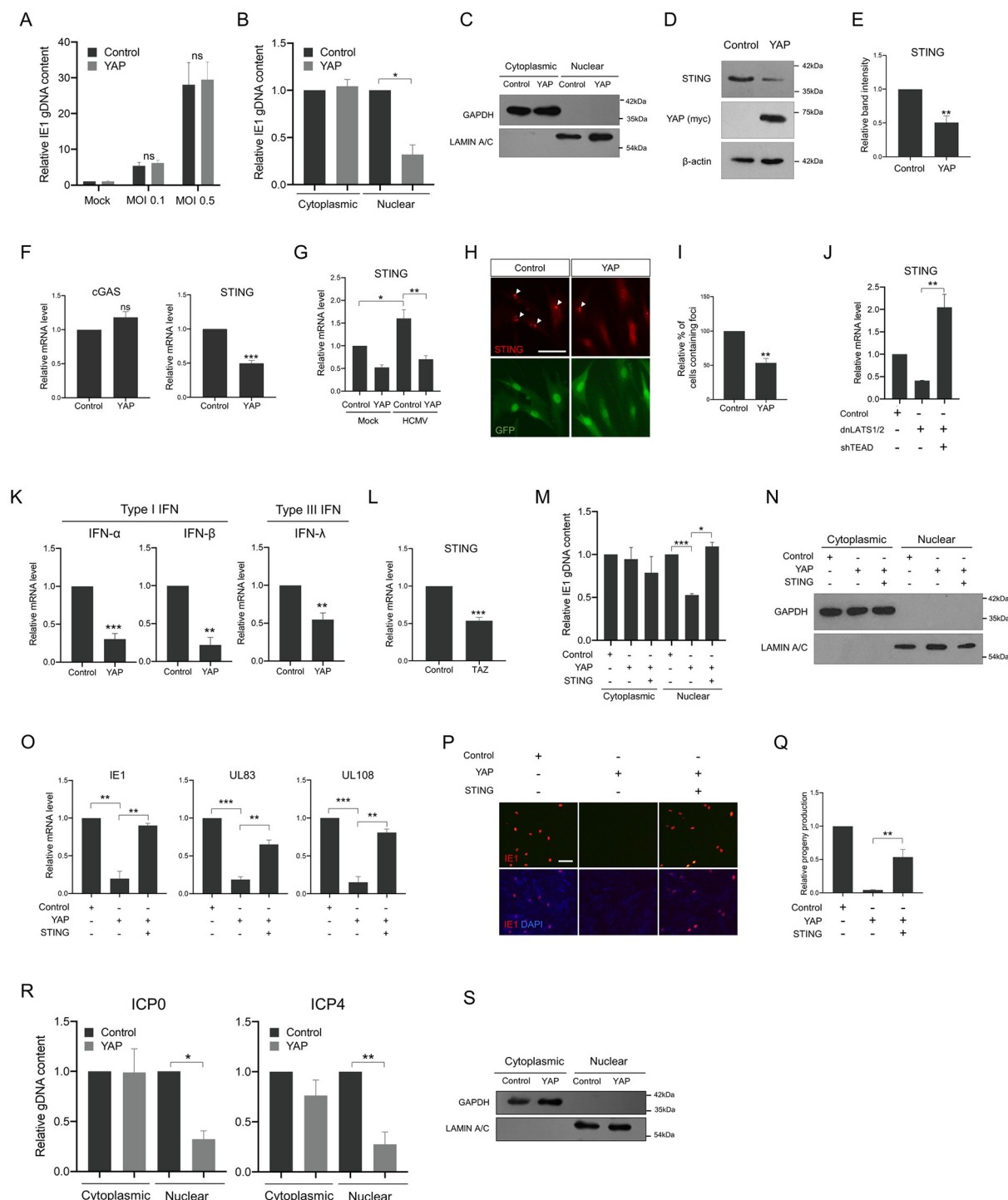

**Fig 4. YAP inhibits nuclear transport of the HCMV genome by repressing STING expression.** (A) HCMV-added HFF cells previously transduced with YAP-expressing retroviral vectors were kept at 4˚C for 1 h and incubated at 37˚C for another 1 h. Non-internalized virions were removed by trypsin and EDTA treatment. Cells were then lysed and relative amounts of internalized virus particles were measured by qPCR of viral genomic IE1. (B) Control or YAP-expressing HFF cells were infected with HCMV at an MOI of 0.5 and harvested at 4 hpi for subcellular fractionation followed by qPCR analysis of genomic HCMV IE1 sequences. qPCR data were normalized to host MDM2 DNA. (C) The quality of fractionation was tested by Western blot detecting GAPDH (cytoplasm) and LAMIN A/C (nucleus) proteins. (D) Western blot, (F) qPCR and (H) immunocytochemical analyses of STING gene expression using YAP-transduced HFF cells. Closed arrow heads in (H) indicate STING foci. (E, I) Quantification of band intensities and number of STING+ foci in (D, H), respectively. (G) YAP-expressing HFF cells were infected with HCMV at an MOI of 0.5, and harvested for qPCR analysis of mRNA levels of STING at 48 hpi. (J) HFF cells were

transduced with dnLATS1/2, with or without TEAD shRNA (shTEAD)-expressing retroviral vectors, and then harvested for qPCR analysis of STING mRNA at 2 days post-transduction. (K) qPCR analysis of interferon (IFN) gene expression using YAP-expressing HFF cells at 2 days post-transduction. (L) Effect of TAZ expression on the mRNA levels of STING. (M) HFF cells transduced with YAP together with or without STING-expressing retroviral vectors were infected with HCMV at an MOI of 0.5 and harvested at 4 hpi for subcellular fractionation followed by qPCR analysis of genomic HCMV IE1 sequences. qPCR data were normalized to host MDM2 DNA. (N) The quality of fractionation was tested by Western blot detecting GAPDH (cytoplasm) and LAMIN A/C (nucleus) proteins. (O) Viral gene expression pattern was examined by qPCR for representative immediate-early (IE1), early (UL83) and late (UL108) genes at the indicated hpi. (P) In parallel, at 72 hpi, progeny viruses were collected and titered by HFF infection followed by IE1 immunostaining (red). Cells were counterstained with DAPI to visualize nuclei (blue). (Q) Quantification of (P). (R) YAP-transduced HFF cells were infected with HSV-1 at an MOI of 5 and harvested at 2 hpi for fractionation followed by qPCR analysis of genomic HSV-1 ICP0 and ICP4 sequences. qPCR data were normalized to host MDM2 DNA. (S) The quality of subcellular fractionation was tested by Western blot detecting GAPDH (cytoplasm) and LAMIN A/C (nucleus) proteins. Scale bars, 100 μm. $n$ = 3 biological replicates for each experiment except for (B, M, $n$ = 4 biological replicates). Error bars represent SEM. Student's $t$-test (for E, F, I, K, L), one-way ANOVA with Turkey's multiple comparisons test (for B, G, J, M, O, Q, R) and two-way ANOVA with Sidak's multiple comparison test (for A) were used to determine statistical significance. $^*P < 0.05$, $^{**}P < 0.01$ and $^{***}P < 0.001$.

nucleus [24], and immunolabeling showed a reduced number of STING foci in YAP-expressing cells regardless of HCMV infection (Fig 4H and 4I, and S1 Fig). The foci were observed to disappear after STING shRNA expression (S1 Fig), validating that the foci structures were indeed composed of STING proteins. STING expression also decreased upon endogenously increased YAP, and TEAD knockdown reversed the effect over control levels, confirming again the importance of TEAD in YAP functions (Fig 4J). As expected as consequences for STING downregulation, mRNA levels of interferon and interferon-stimulated genes (ISGs) were reduced by YAP (Fig 4K and S2 Fig), and these results suggest that YAP can modulate host immune responses through regulation of STING expression.

Furthermore, TAZ was also shown to have the ability to inhibit STING expression (Fig 4L), suggesting that the structurally closely related YAP and TAZ proteins can interfere with HCMV replication through a similar molecular mechanism. Although not the subject of this study, we also examined whether YAP could target cGAS expression, since YAP was already known to negatively regulate TBK1 and IRF3 in the cGAS/STING/TBK1/IRF3 cascade [25,26], and found that cGAS expression was not affected by YAP (Fig 4F).

The functional importance of reduced STING expression in YAP-suppressed HCMV replication was examined by qPCR to assess viral genome nuclear transport and HCMV viral gene expression patterns. As shown in Fig 4M and 4N, co-expression of STING reverted the YAP-impaired delivery of the HCMV genome into the nucleus, and subsequently, the YAP-attenuated mRNA expression of immediate-early, early and late HCMV genes was restored by STING co-expression (Fig 4O). A viral progeny assay also showed that YAP-reduced infectious HCMV progeny production was rescued upon STING expression (Fig 4P and 4Q). Given the report that herpes simplex virus type 1 (HSV-1) also requires STING for efficient nuclear viral genome delivery [23], we tested the effects of YAP, and found that YAP expression significantly inhibited the nuclear import of HSV-1 genome, like the HCMV genome (Fig 4R and 4S). Taken together, these data strongly indicate that YAP-inhibited HCMV replication is attributable to the reduced expression of STING.

## HCMV infection induces YAP expression to counteract HCMV-triggered STING expression

So far, our study has focused on the role of YAP in the nuclear transport of HCMV, a step long before viral gene expression. We attempted to extend the scope of the study to post-infection stages by performing qPCR to analyze YAP and STING expression patterns after HCMV infection. We found that, overall, both YAP and STING mRNA levels were elevated during the early stages of the HCMV life cycle (Fig 5A and 5B) and confirmed the HCMV-induced YAP and STING expression at the protein level by Western blot analysis (Fig 5C–5E). Particularly,

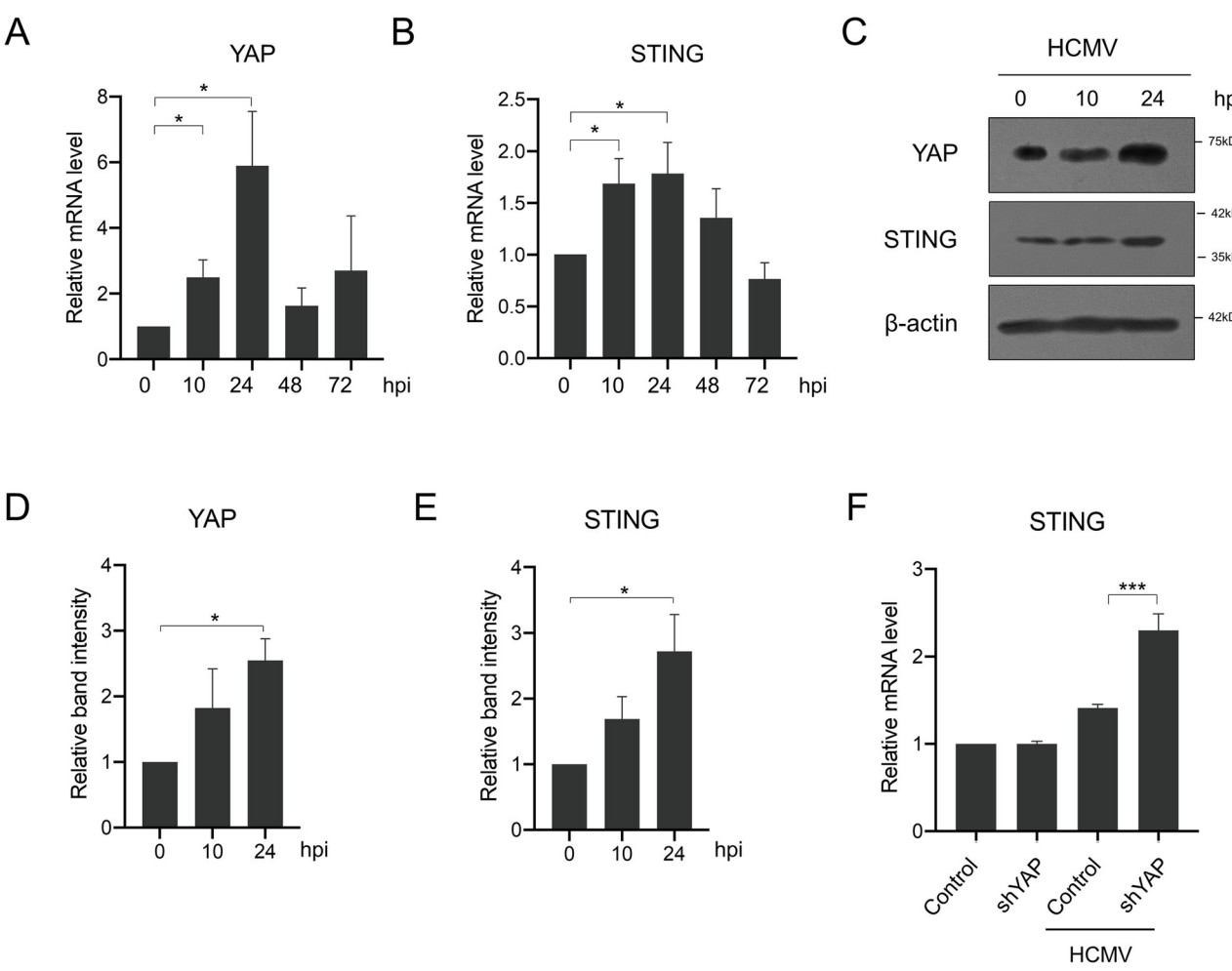

**Fig 5. HCMV infection increases YAP at the transcriptional level to regulate STING gene expression.** qPCR analysis of (A) YAP and (B) STING gene expression was performed using HFF cells infected with HCMV at an MOI of 0.5 and harvested at the indicated hpi. (C) Western blot analysis of YAP and STING proteins using HFF cells infected with HCMV at an MOI of 0.5. (D, E) Quantification of YAP and STING band intensities, respectively, in (C). (F) YAP shRNA-expressing HFF cells infected with HCMV at an MOI of 0.5 were collected for qPCR analysis of STING. $n = 3$ biological replicates for each experiment. Error bars represent SEM. One-way ANOVA with Turkey's multiple comparisons test was used to determine statistical significance. $^*P < 0.05$ and $^{***}P < 0.001$.

YAP mRNA increase was already evident at 10 hpi and peaked at 24 hpi (Fig 5A). Transcription of STING was also enhanced (Fig 5B), presumed to be attributable to a cellular event involving host antiviral immune mechanisms [27,28]. From these results, we inferred that HCMV-increased YAP may be used to curb STING mRNA production as a viral counter-regulatory response against the host innate immune system. Indeed, qPCR analysis revealed that HCMV-triggered STING mRNA expression was further enhanced in YAP-reduced conditions (Fig 5F), indicating that HCMV-induced YAP effectively minimizes the increase of STING expression. Taken together, these data suggest that HCMV can negatively regulate STING functions by increasing YAP.

## Discussion

In this study, we demonstrated that YAP-suppressed STING expression leads to impairment of HCMV replication from the very early stages of HCMV infection. Since YAP lacks DNA-

binding activity, we expected that a YAP-interacting coactivator would mediate the actions of YAP, and subsequently found that YAP repression of HCMV virion production and STING expression is TEAD-dependent. YAP is known to show transcriptional activator or repressor activity depending on the type of partner transcription factor, and is known to act as a transcriptional repressor in two ways. First, the nucleosome remodeling and histone deacetylase (NuRD) complex was proposed to mediate the repressor activity of YAP [29]. Second, interaction with Yin Yang (YY1) and enhancer of zeste homologue 2 (EZH2) was shown to induce suppression of YAP target gene expression [30]. However, we found no consensus TEAD binding sequences in the promoter-proximal regions of the STING gene, supporting the possibility that YAP suppresses STING expression in an indirect manner by regulating transcription of an unknown activator or repressor for STING expression.

Since both YAP and TAZ are mammalian homologs of *Drosophila* Yorkie in the Hippo signaling pathway [12,31] and share high sequence similarity, they have been thought to perform overlapping functions. However, there are also significant distinctions between YAP and TAZ [32–34]. In terms of biochemical properties, YAP proteins appear in a diffused staining pattern in the nucleus, whereas TAZ proteins localize to punctate nuclear bodies [32]. In addition, TAZ lacks a proline-rich region, therefore, as the origin of its name suggests, only YAP can bind to YES through its proline-rich domain. In this study, we found that YAP requires the TEAD cofactors for inhibiting HCMV replication and expected that TAZ expression would exhibit similar effects since TAZ also contains a TEAD binding domain. Indeed, TAZ showed similar inhibitory activities in all experimental settings used in this study. However, these results do not necessarily suggest that the viral replication-related functions of YAP and TAZ are redundant in real-life situations. The important difference between YAP and TAZ may arise from distinct tissue expression patterns rather than their different biochemical properties. YAP is highly expressed in the lung, ovary and testis, whereas TAZ mRNA is found at high levels in the kidney [35]. Therefore, YAP and TAZ are complementary rather than overlapping. Currently, many research groups are mainly focusing on YAP, but depending on the host cell and tissue types, additional attention is needed to investigate the role of TAZ in diverse viral pathogenesis.

The key findings in this study are that YAP-inhibited STING expression results in impaired viral genome delivery and reduced interferon production. These observations can be interpreted to mean that YAP prevents excessive multiple viral infections which could cause damage to host cells or break latency, and that YAP thus enables HCMV to avoid immune attacks. Putting these together in terms of HCMV infection, we hypothesize that HCMV may use YAP to stably maintain the infection state by keeping a low profile without destroying host cells, while evading the antiviral immune response.

So far, most studies elucidating the relationship between YAP and viral pathogenesis have focused on changes in the amount of YAP protein after viral infection (infection > YAP). Some reports describe the effects of YAP on viral infections (YAP > infection), but most of their subjects were YAP-induced immune evasion mechanisms reducing antiviral cytokine production [25], thus affecting diverse viral infections non-specifically and indirectly, rather than each virus-specific effects of YAP. In this regard, a previous report showing that YAP induced EBV immediate early promoter activation promoting viral lytic reactivation in epithelial cells [36] is noteworthy. Our study demonstrating YAP-inhibited HCMV replication by modulation of cellular nuclear transport machinery is also unique in two respects in that it addresses (1) a herpes virus-specific role of YAP (2) independent of host innate immune response modulation.

Before discussing the potential role of YAP in other viral infections as an upstream factor of STING, it is important to first note the features of STING. STING has been studied as a

negative host factor for viral infections since it is a key component of the cGAS/STING/TBK1/IRF3 regulatory axis activating interferon pathway [37,38], and therefore as a target of viral immune evasion. However, recent emerging studies suggest that STING can regulate viral replication in a host immune response-independent manner, and that this non-canonical activity of STING can function positively for some viral replications. For example, STING enhances replication and transmission of human rhinovirus (HRV) via autophagy [39]. STING-mediated HCMV viral DNA transport, which is the topic of this study, is also a non-canonical function of STING. Since STING plays an important role in many viral replications in diverse ways as described, our study identifying YAP as an upstream modulator of STING provides insight into the potential involvement of YAP in a variety of viral pathogeneses. In this regard, our data showing YAP-reduced nuclear import of the HSV-1 genome provides strong support for this idea.

## Materials and methods

### Plasmids

The wild-type YAP vector was purchased from Addgene (plasmid #33091, Cambridge, MA) and subcloned into the MluI site of the retroviral vector MSIG [40]. The YAP ΔPDZ, S94A, WW1* and WW2* mutants were generated by site-directed mutagenesis as described previously [41]. The STING gene was amplified using the primers, STING forward (5'-TTCTGTATTTGTCTGAAAATAGCCACCATGCCCCACTCCAGCCTGCATCC-3') and STING reverse (5'-CTCTCCGCACGGATTTCTCTGAACAAAAACTCATCTCAGAAGAGGATCTGTGATGGGCCCGCGGATCCAATTC-3'), and inserted into MSIG by Gibson assembly reaction (New England Biolabs, Beverly, MA). For knockdown experiments, the STING shRNA plasmids were purchased from Sigma (TRCN 0000161052, Saint-Louis, MO). The YAP and TEAD shRNA vectors are kind gifts from Dr. Kun-Liang Guan (University of California San Diego).

### Retroviral vector production

The retroviral vector production protocol has previously been described [41]. Briefly, gag-pol (pCA-gag-pol) and env-expressing vector (vesicular stomatitis virus glycoprotein) were transfected with the retroviral construct into human embryonic kidney (HEK) 293T cells using polyethyleneimine (Sigma). After 48 h of transfection, the supernatant was collected and filtered through a 0.45 μm filter. Viral stock concentration was conducted by ultracentrifugation at 25,000 rpm for 90 min at 4°C in an SW28 rotor (Beckman Coulter, Fullerton, CA). Pellets were resuspended in 100 μl of phosphate-buffered saline (PBS) at 4°C for about 12 h, and aliquoted virus stocks were stored at -80°C.

### HCMV progeny assay

Before HCMV (Towne strain) infection, human foreskin fibroblast (HFF) cells were transduced with a retroviral vector expressing YAP and cultured for 24 h. Cells were then incubated with viral inocula for 1 h at a multiplicity of infection (MOI) of 0.1 or 0.5. Cell culture supernatants and cell debris dissociated by repeated freezing and thawing three times were collected from 72 to 120 h post-infection (hpi) and centrifuged at 14,000 rpm for 10 min at 4°C. For viral titration, untransduced HFF cells were incubated with the harvested HCMV for 1 h at 37°C. At 48 hpi, viral concentrations were determined by immunofluorescence assays using an anti-IE1 antibody as described below.

## HCMV cell entry assay

HCMV-challenged cells were incubated for 1 h at 4°C and an additional 1 h at 37°C, and non-entered virus was removed by trypsin (0.025%)/EDTA (0.01%) solution (Thermo Fisher Scientific, Waltham, MA). Cells were then treated with lysis buffer containing proteinase K (100 μg/ml) overnight at 55°C. The lysate was centrifugated at 13,000 rpm for 5 min and HCMV DNA was purified from supernatants by ethanol precipitation. Relative amounts of HCMV viral genome were further analyzed by quantitative real-time PCR detecting IE1 DNA using the primers, HCMV IE1 forward (5'-CAAGTGACCGACCATTGCAA-3') and HCMV IE1 reverse (5'-CACCATGTCCACTCGAACCTT-3').

## Quantitative real-time PCR (qPCR)

Total RNA was prepared from HCMV-infected cells using TRIzol reagent (Takara, Otsu, Japan) and cDNAs were synthesized from 500 ng of each RNA sample using an oligo (dT) primer and ReverTraAce reverse transcriptase (Toyobo, Osaka, Japan). qPCR for HCMV gene expression was performed according to the Smart Cycler System protocol (Takara) using the following primers: IE1 forward (5'-CAAGTGACCGAGGATTGCAA-3'), IE1 reverse (5'-CACCATGTCCACTCGAACCTT-3'), UL44 forward (5'-GCTGTCGCTCTCCTCTTTCG-3'), UL44 reverse (5'-TCACGGTCTTTCCTCCAAGG-3'), UL99 forward (5'-GTGTCCCATTCCCGACTCG-3'), UL99 reverse (5'-TTCACAACGTCCACCCACC-3'), UL83 forward (5'-GCAGAACCAGTGGAAAGAGC-3'), UL83 reverse (5'-GTCCTCTTCCACGTCAGAGC-3'), UL108 forward (5'-TCTGGCTCGACACAATGATCAC-3'), UL108 reverse (5'-GCTAATTGGACTTTGCCCATGT-3'), β-actin forward (5'-GGCCAACCGCGAGAAGATGA-3'), β-actin reverse (5'-CCAGAGGCGTACAGGGATAG-3'). For the qPCR of YAP, CTGF, STING, cGAS, interferon, ISG genes, the following primers were used: YAP forward (5'-TAGCCCTGCGTAGCCAGTTA-3'), YAP reverse (5'-TCATGCTTAGTCCACTGTCTGT-3'), STING forward (5'-CCTGAGTCTCAGAACAACTGC-3'). STING reverse (5'-GGTCTTCAAGCTGCCCACAGT-3'), cGAS forward (5'-GGGAGCCCTGCTGTAACACTTCTTAT-3'), cGAS reverse (5'-CCTTTGCATGCTTGGGTACAAGGT-3'), CTGF forward (5'-ACCGACTGGAAGACACGTTTG-3'), CTGF reverse (5'-CCAGGTCAGCTTCGCAAGG-3'), IFNalpha forward (5'-TCGCCCTTTGCTTTACTGAT-3'), IFNalpha reverse (5'-GGGTCTCAGGGAGATCACAG-3'), IFNbeta forward (5'-AAACTCATAGCAGTCTGCA-3'), IFNbeta reverse (5'-AGGAGATCTTCAGTTTCGGAGG-3'), IFNlambda forward (5'-CCAGTGATGATTCTCTTGAGAGC-3'), IFNlambda reverse (5'-CCCCAAAGCGTAGAGGTCCA-3'), ISG 12 forward (5'-AATCGCCTCGTCCTCCATAGCA-3'), ISG 12 reverse (5'-CCTCGCAATGACAGCCGCAAT-3'), ISG 15 forward (5'-ATGGGCTGGGACCTGACG-3'), ISG15 reverse (5'-GCCAATCTTCTGGGTGATCTG-3'), ISG16 forward (5'-CTCGCTGATGAGCTGGTCT-3'), ISG16 reverse (5'-ATACTTGTGGGTGGCGTAGC-3').

## Western blot assay

HFF cells were lysed with radioimmunoprecipitation assay (RIPA) buffer (Sigma) containing a protease and phosphatase inhibitor cocktail (Pierce Biotechnology, Waltham, MA). Equal amounts of protein (20 to 40 μg, depending on the target protein) were resolved in 8 to 10% (wt/vol) SDS-PAGE and transferred to polyvinylidene fluoride (PVDF) membranes (PALL, Cortland, NY). The membranes were blocked with TBST (150 mM NaCl, 10 mM Tris-HCl, 0.1% [vol/vol] Tween 20, pH 8.0) containing 5% (wt/vol) skim milk and analyzed with the following primary antibodies: anti-YAP (1:1000, Santa Cruz Biotechnology, Santa Cruz, CA, sc-101199), anti-STING (1:500, R&D systems, Minneapolis, MN, MAB7169), anti-IE1 (1:1000,

Millipore, Billerica, MA, MAB8131), anti-pp52 (1:1000, Virusys, Sykesville, MD, ICP36), anti-pp28 (1:1000, Santa Cruz Biotechnology, sc-69749), and anti-β-actin (1,10,000, Santa Cruz Biotechnology, sc-47778). All blots were incubated overnight with primary antibody at 4˚C with horseradish peroxidase-conjugated anti-mouse or anti-rabbit secondary antibodies for 2 h at room temperature. The enhanced chemiluminescence system (Atto, Tokyo, Japan) and X-Omat film (Kodak, Rochester, NY) were used to visualize the protein bands.

### Immunofluorescence staining

Standard immunofluorescence procedures were used for visualization of IE1 and STING proteins in HCMV-infected cells. Briefly, cells were fixed with 4% paraformaldehyde for 15 min, permeabilized with 100% cold ethanol for 1 min 30 sec and washed with PBS three times. Cells were then incubated overnight at 4˚C with anti-GFP (1:1000, Abcam, Cambridge, UK, ab13970), anti-IE1 (1:2000, Millipore, MAB8131), or anti-STING antibody (1:1000, R&D systems, MAB7169), washed three times in PBS and incubated for 2 h at room temperature with Alexa-555-conjugated secondary antibody (1:1,000, Invitrogen, Carlsbad, CA) diluted in PBS. DAPI staining was used to visualize cell nuclei. Fluorescent images were obtained with an inverted microscope (Eclipse Ti, Nikon, Tokyo, Japan).

### Subcellular fractionation for viral nuclear transport assay

HFF cells were infected with HCMV at an MOI of 0.5 and harvested at 4 hpi. Subcellular fractionation was then performed using a kit purchased from Thermo Fisher Scientific (#78833) according to the manufacturer's instructions. HCMV genomic DNA from each fraction was extracted using the Expin PCR SV kit (GeneAll, Seoul, South Korea, #103–102). Relative amounts of HCMV viral genome were analyzed by qPCR detecting IE1 using the following primers: HCMV IE1 forward (5'-CAAGTGACCGACCATTGCAA-3') and IE1 reverse (5'-CACCATGTCCACTCGAACCTT-3'), For the HSV-1 nuclear transport assay, HFF cells were infected with HSV-1 at an MOI of 5 and collected at 2 hpi for fractionation. Relative amounts of the HSV-1 viral genome were analyzed by qPCR detecting the ICP0 and ICP4 genes using the following primers: HSV-1 ICP0 forward (5'-CAAGTGACCGACCA TTGCAA-3') and HSV-1 ICP0 reverse (5'-CACCATGTCCACTCGAACCTT-3'), HSV-1 ICP4 forward (5′- CAGAGCCAAGCGGCGGCAGA -3′), and HSV-1 ICP4 reverse (5′-AGAAGCTGCTGGTGGCGGGG-3′). Amplified viral sequences were normalized by cellular MDM2 DNA content using MDM2 forward (5′- CAGAGCCAAGCGGCGGCAGA -3′) and MDM2 reverse (5′- AGAAGCTGCTGGTGGCGGGG-3′) primers and the quality of fractionation was examined by Western blot using anti-GAPDH (cytoplasm: 1:1000, Cell Signaling Technology, Danvers, MA, #5174S) and anti-LAMIN A/C (nucleus: 1:1000, Cell Signaling Technology, #2032S).

### Statistical analysis

Statistical tests were performed using Prism 8 software (GraphPad). Student's two-tailed *t*-test, and one- or two-way ANOVA were performed for statistical analyses. All data represent three or more independent experiments.

## Supporting information

**S1 Data. Excel spreadsheet containing, in separate sheets, the underlying numerical data and statistical analysis for Figs 1D, 1F, 1H, 1J, 2A, 2C, 3B, 3D, 3F, 3H, 3J, 4A, 4B, 4E, 4F,**

**4G, 4I, 4J, 4K, 4L, 4M, 4O, 4Q, 4R, 5A, 5B, 5D, 5E and 5F, S1B, S1D, S1F** and **S2** Figs.
(XLSX)

**S1 Fig. Validation of STING foci.** (A) Western blot and (C) immunocytochemical analyses of STING proteins using STING shRNA-transduced HFF cells at 2 days post-transduction. Closed arrow heads in (C) indicate STING foci. (E) YAP-transduced HFF cells infected with HCMV were immunostained using anti-STING antibody (red). (B, D, F) Quantification of band intensities and the number of STING⁺ foci in (A, C, E), respectively. Scale bars, 100 μm. $n = 3$ biological replicates for each experiment. Error bars represent SEM. Student's $t$-test was used to determine statistical significance. $^{***}P < 0.001$.
(TIF)

**S2 Fig. YAP reduces gene expression of interferon-stimulated genes.** qPCR analysis of interferon-stimulated genes (ISGs) were performed using YAP-expressing HFF cells at 2 days post-transduciton. $n = 3$ biological replicates for each experiment. Error bars represent SEM. Student's $t$-test was used to determine statistical significance. $^{***}P < 0.001$.
(TIF)

**S3 Fig. Uncropped gel images for Figs 1G, 2B, 4C, 4D, 4N, 4S and 5C, and S1A Fig.**
(PDF)

## Author Contributions

**Conceptualization:** Ju Hyun Lee, Dasol Han, Keejung Yoon.

**Data curation:** Ju Hyun Lee, Keejung Yoon.

**Formal analysis:** Ju Hyun Lee, Mookwang Kwon, Woo Young Lim, Chae Rin Yoo, Youngik Yoon, Dasol Han.

**Funding acquisition:** Keejung Yoon.

**Investigation:** Ju Hyun Lee, Mookwang Kwon, Woo Young Lim, Chae Rin Yoo, Youngik Yoon, Dasol Han.

**Methodology:** Ju Hyun Lee, Keejung Yoon.

**Project administration:** Keejung Yoon.

**Resources:** Jin-Hyun Ahn, Keejung Yoon.

**Supervision:** Keejung Yoon.

**Validation:** Ju Hyun Lee, Mookwang Kwon, Woo Young Lim.

**Visualization:** Ju Hyun Lee, Dasol Han.

**Writing – original draft:** Ju Hyun Lee, Keejung Yoon.

**Writing – review & editing:** Ju Hyun Lee, Jin-Hyun Ahn, Keejung Yoon.

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
