## [Decision Letter · Decision Letter 0]

10 Aug 2022

Dear Professor Yoon,

Thank you very much for submitting your manuscript "YAP inhibits HCMV replication by impairing STING-mediated nuclear transport of the viral genome" for consideration at PLOS Pathogens. As with all papers reviewed by the journal, your manuscript was reviewed by members of the editorial board and by several independent reviewers. In light of the reviews (below this email), we would like to invite the resubmission of a significantly-revised version that takes into account the reviewers' comments.

The reviewers find the basic observations regarding the interaction of YAP and STING in the context of HCMV replication intriguing but are generally in agreement about the manuscript's deficiencies with regard to an understanding of the molecular mechanisms of these relationships. In addition, concerns about the nature of the experimental systems and lack of orthogonal methods use were also mentioned.

We cannot make any decision about publication until we have seen the revised manuscript and your response to the reviewers' comments. Your revised manuscript is also likely to be sent to reviewers for further evaluation.

Sincerely,

Victor Robert DeFilippis

Guest Editor

PLOS Pathogens

Klaus Früh

Section Editor

PLOS Pathogens

Kasturi Haldar

Editor-in-Chief

PLOS Pathogens

orcid.org/0000-0001-5065-158X

Michael Malim

Editor-in-Chief

PLOS Pathogens

orcid.org/0000-0002-7699-2064

The reviewers find the basic observations regarding the interaction of YAP and STING in the context of HCMV replication intriguing but are generally in agreement about the manuscript's deficiencies with regard to an understanding of the molecular mechanisms of these relationships. In addition, concerns about the nature of the experimental systems and lack of orthogonal methods use were also mentioned.

Reviewer's Responses to Questions

**Part I - Summary**

Reviewer #1: In this paper, the authors revealed that YAP, well known for its involvement in the mammalian Hippo signaling pathway, inhibits HCMV replication by STING-mediated nuclear delivery of the viral genome. The authors observed that YAP inhibits the expression of STING at least at the transcription level, and showed that overexpression of STING is able to restore YAP-attenuated HCMV replication. STING is a well-known nucleic acid sensor that has long been known to activate the anti-viral interferon responses, and has recently been shown to be a pro-viral protein that promotes the nuclear delivery of herpesvirus genomes. Based on this previous report, the authors concluded that YAP inhibits the expression of STING, and consequently the nuclear transfer of HCMV genome is impaired, thereby inhibiting infection. It is a very interesting result, but the data is still premature to support the authors' conclusions. Further in-depth experiments are needed to reinforce the authors' conclusions. In particular, it is necessary to delineate what molecular mechanism YAP, the core subject of this paper, regulates the expression of STING.

Reviewer #2: In this manuscript, Lee and colleagues examine the role of the Hippo signaling pathway and YAP transcription factor in viral infection, namely the b-herpesvirus HCMV. They find that YAP is an inhibitor of early HCMV life cycle, by inhibiting STING-dependent trafficking of the viral genome. In general, this manuscript details a new and potentially interesting link between Hippo signaling/YAP, STING expression and HCMV replication. However, further control and mechanistic experiments are required in order to fully support and strengthen the proposed hypothesis, including a physiological method of examining the effects of YAP on HCMV replication apart from over-expression. Further, the established role of STING in anti-viral cellular immunity and cell viability after viral infection must be addressed. Finally, controls confirming the validity of STING antibody staining are necessary.

Major Points

1. In Figure 1 over-expression experiments, what are the wt and over-expression levels of YAP in cells? Is the degree of YAP over-expression physiological, in terms of relation to wt levels? Does attenuation of endogenous YAP expression have a discernible effect on HCMV replication? The control vector should ideally express an inert protein, because ectopic protein expression in itself could inhibit viral replication.

2. Does YAP over-expression alone adversely affect intrinsic cellular viability or replication, which would inhibit HCMV viral progeny? What about the YAP mutants?

3. Protein levels in Figure 2B (western blot) would be most useful if quantified.

4. Does YAP expression have an effect on levels of anti-viral type I or type III IFN cytokines, before or after expression?

5. The authors state that STING typically localizes to the nucleus, however in unstimualted cells STING typically localizes to ER membrane. After stimulation with DNA or cyclic-di-nucleotides, STING typically re-localizes to ER-adjacent peri-nuclear structures, and co-localizes with TBK1. It is unclear what the structures in Figure 5E correspond to, as they appear to be detectable in unstimulated cells. Can the authors confirm these structures actually correspond to STING protein? If STING level are drastically different, why is the overall STING intensity not different comparing the control and YAP expressing cells in Figure 5E (e.g. staining outside the foci)?

6. Does STING directly associate with HCMV viral genome sequences in HFF?

7. How do the authors rationalize lower STING levels (and presumably lower IFN levels) and lower HCMV replication? Does the effect of YAP over-expression change later in infection, due to paracrine effects of IFN?

Minor Points

1. Not all of the data panels are properly discussed in the results section (e.g. Fig 1C and 1E)

2. In Figure 1B and 1D, the differences between the panels (e.g. MOI) should be clearly indicated.

Reviewer #3: This paper by Lee et al. describes that YAP which is a critical regulator of the Hippo signaling pathway is able to negatively affect HCMV replication at the level of viral genome nuclear import. Essentially, the authors demonstrate that overexpresson of YAP inhibits HCMV replication. Further experiments show that this happens latest at the stage of immediate early gene expression and requires the transcriptional activation function of YAP. As an underlying mechanism it is shown that YAP downregulates STING and since STING (as shown in a previous publication) is required for the nuclear import of viral DNA, inhibition of the import of viral DNA into the nucleus accounts for the negative effect of YAP on viral replication.

This is a rather short paper, however, it is well structured and most results are convincing. One major criticism, however, relates to the fact that all results were obtained upon conditions of strong overexpression of YAP in primary human fibroblasts.

Thus, the major queston that needs to be answered by the authors is: does YAP indeed affect HCMV replication in the context of physiologically relevant conditions? The authors need to provide data concerning endogenous YAP levels to identify cell types that express high levels of YAP. Kockdown and infection experiments should then be used to show that YAP mediated inhibition of nuclear import is indeed a physiologically relevant mechanism that affects HCMV replication. Furthermore, it would be interesting to see whether YAP also affects other herpesviruses (e.g. herpes simplex virus).

**Part II – Major Issues: Key Experiments Required for Acceptance**

Reviewer #1: 1. What are the dynamics of spatio-temporal expression of YAP and STING expression during HCMV infection?

1) Authors should analyze the expression of YAP and STING during the immediate-early stage of HCMV infection. HCMV infection appears to upregulate mRNA level of STING. What about changes in STING protein levels? Data show that overexpression of YAP inhibits the appearance of STING foci. Is a similar phenomenon observed in the context of HCMV infection?

2) YAP protein is a core protein in Hippo signaling which regulates cell proliferation/death. Cell proliferation could mislead the quantification of the amount of DNA transferred into the nucleus. Therefore, it is important to rule out the possibility that the ectopic expression of YAP protein did not affect cell proliferation.

3) To clarify YAP represses HCMV viral gene expression through the regulation of STING expression, authors are encouraged to show that the Hippo signaling pathway is not activated during HCMV infection. It can be done by phosphor-immunoblotting for the components of Hippo signaling pathway components such as Hippo, MST1/2 and LATS1/2. Given the previous reports that the activity of YAP depends on its phosphorylation, it may provide valuable information to the direction of in-depth research to see if there is a correlation between the phosphorylation status of YAP and STING expression during productive HCMV infection?

2. What are the molecular mechanisms by which YAP controls the cellular level of STING? Some specific questions are as follows.

1) Does HCMV manipulate the expression of STING through YAP related pathway?

2) Does HCMV infection affect the expression of endogenous YAP protein?

3) Does YAP directly activate the STING promoter?

4) YAP functions as transcription factor by interacting with multiple co-activators such as TEAD family, Runx, and AP2. Although authors confirm that interaction between YAP and TEAD family is required for the function of YAP, it should be more thoroughly validated.

5) The gain-of-function study must be accompanied by a loss-of-function study to prove that it is not an artificial outcome. Overexpression of YAP decreased HCMV replication. Does depletion of YAP enhance HCMV replication? Try knockdown or knockout experiments of YAP and proceed to assay viral growth (RNA, protein level, and titration of progeny virus)

Reviewer #2: 1. Endogenous YAP knockout or knockdown (loss of function) to validate over-expression results

2. Cell viability after YAP over-expression

3. Validate STING protein staining for immunofluorescence experiments (Fig 5E)

Reviewer #3: 1. The authors need to identify cell types and/or conditions where YAP is expressed at high levels. Knockdown experiments should then be performed in order to demonstrate that YAP indeed inhibits HCMV replication in a physiological context.

2. The authors should use other herpesviruses (e.g. HSV-1, VZV) as further controls in their experiments.

**Part III – Minor Issues: Editorial and Data Presentation Modifications**

Reviewer #1: Several overlapping and redundant data with the same conclusion are presented in this manuscript. Figures 1 and 2 can be merged. Figures 4, 5, and 6 repeat the same information and must be merged together.

Reviewer #2: 1. Not all of the data panels are properly discussed in the results section (e.g. Fig 1C and 1E)

Reviewer #3: 1. The authors need to indicate in each figure legend the number of biological replicates that was used to calculate the statistical significance.

PLOS authors have the option to publish the peer review history of their article (what does this mean?). If published, this will include your full peer review and any attached files.

Reviewer #1: No

Reviewer #2: **Yes: **Sonia Sharma

Reviewer #3: No
---

## [Decision Letter · Decision Letter 1]

17 Nov 2022

Dear Professor Yoon,

We are pleased to inform you that your manuscript 'YAP inhibits HCMV replication by impairing STING-mediated nuclear transport of the viral genome' has been provisionally accepted for publication in PLOS Pathogens.

Best regards,

Victor Robert DeFilippis

Guest Editor

PLOS Pathogens

Klaus Früh

Section Editor

PLOS Pathogens

Kasturi Haldar

Editor-in-Chief

PLOS Pathogens

orcid.org/0000-0001-5065-158X

Michael Malim

Editor-in-Chief

PLOS Pathogens

orcid.org/0000-0002-7699-2064

Reviewer Comments (if any, and for reference):

Reviewer's Responses to Questions

**Part I - Summary**

Reviewer #1: The revision was properly made as requested by this reviewer. There is no further comment.

Reviewer #3: The authors have addressed all major concerns raised by this reviewer. The provided data support the conclusion that YAP inhibits HCMV and also HSV replication via affecting STING-mediated transport of viral genomes.

**Part II – Major Issues: Key Experiments Required for Acceptance**

Reviewer #1: The revision was properly made as requested by this reviewer. There is no further comment.

Reviewer #3: None

**Part III – Minor Issues: Editorial and Data Presentation Modifications**

Reviewer #1: The revision was properly made as requested by this reviewer. There is no further comment.

Reviewer #3: None

PLOS authors have the option to publish the peer review history of their article (what does this mean?). If published, this will include your full peer review and any attached files.

Reviewer #1: No

Reviewer #3: No

---

## [Editor Report · Acceptance letter]

28 Nov 2022

Dear Professor Yoon,

We are delighted to inform you that your manuscript, "YAP inhibits HCMV replication by impairing STING-mediated nuclear transport of the viral genome," has been formally accepted for publication in PLOS Pathogens.

Best regards,

Kasturi Haldar

Editor-in-Chief

PLOS Pathogens

orcid.org/0000-0001-5065-158X

Michael Malim

Editor-in-Chief

PLOS Pathogens

orcid.org/0000-0002-7699-2064